# Laplace Approximated Gaussian Process State-Space Models

**Jakob Lindinger**[1,2,3]         **Barbara Rakitsch**[1]         **Christoph Lippert**[2,3]

[1]Bosch Center for Artificial Intelligence, Renningen, Germany
[2]Hasso Plattner Institute, Potsdam, Germany
[3]University of Potsdam, Germany

## Abstract

Gaussian process state-space models describe time series data in a probabilistic and non-parametric manner using a Gaussian process transition function. As inference is intractable, recent methods use variational inference and either rely on simplifying independence assumptions on the approximate posterior or learn the temporal states iteratively. The latter hampers optimization since the posterior over the presence can only be learned once the posterior governing the past has converged. We present a novel inference scheme that applies stochastic variational inference for the Gaussian process posterior and the Laplace approximation on the temporal states. This approach respects the conditional dependencies in the model and, through the Laplace approximation, treats the temporal states jointly, thereby avoiding their sequential learning. Our method is computationally efficient and leads to better calibrated predictions compared to state-of-the art alternatives on synthetic data and on a range of benchmark datasets.

## 1 INTRODUCTION

Uncertainty estimation in time-series modeling [see e.g. Särkkä, 2013] is a hard task since two different noise sources have to be taken into account: First, the observation noise that stems from possibly noisy measurements of the system under consideration and second, the process noise associated with the uncertain development of the system. Especially the second noise source leads to an accumulation of uncertainty over time and renders predictions far into the future difficult.

Gaussian processes [Rasmussen and Williams, 2006] provide a well established framework for dealing with uncertainty. They define a distribution over functions $f \sim p(f)$ and can be used as building blocks in state-space models for

modeling complex temporal dependencies. This model class, aptly called Gaussian process state-space models [Frigola, 2015], can be used to model observations $y_t \in \mathbb{R}^{d_y}$ from a time series, where $t = 1, \ldots, T$ is the time index. It assumes that each observation $y_t$ is emitted by a latent state $x_t \in \mathbb{R}^{d_x}$, $y_t \sim p(y_t|x_t)$, and that the latent states have a Markovian structure, i.e., $x_{t+1} \sim p(x_{t+1}|x_t, f)$. The noise of this so-called transition model and the function uncertainty of the Gaussian process are propagated over time, resulting in complex and non-Gaussian behavior. The flexibility of this model class paired with its probabilistic predictions makes it an interesting building block e.g. for model-based reinforcement learning [e.g Deisenroth and Rasmussen, 2011].

Since the first work on Gaussian process state space models [Wang et al., 2005], much work has been devoted to deriving efficient and accurate inference schemes. The arguably most expressive model at this point is by Ialongo et al. [2019] which assumes a flexible, parametric Markov-structured Gaussian posterior over the temporal states, $x_t \sim q(x_t|x_{t-1}, f)$, and allows for dependencies with the Gaussian process posterior. However, its empirical performance often does not match its theoretical expressiveness: In the original publication the authors report many cases in which this model is outperformed by an easier alternative that simply sets $q(x_t|x_{t-1}, f)$ to the prior transition [Doerr et al., 2018]. The gap between the empirical and theoretical performance of Ialongo et al. [2019] can most likely be attributed to the hard learning problem created by employing a flexible, parametric $q(x_t|x_{t-1}, f)$: These additional parameters have to be estimated, and, moreover, one can only start optimizing the parameters governing the temporal state $x_t$ once the parameters governing $x_{t-1}$ begin converging since the inference scheme is built on sequential sampling of the temporal states.

In this work we address these issues by presenting a novel inference algorithm for Gaussian process state-space models. Our approach applies stochastic variational inference over the Gaussian process posterior [Hensman et al., 2013] and, conditioned on it, the Laplace approximation [see e.g.

*Accepted for the 38th Conference on Uncertainty in Artificial Intelligence* (UAI 2022).

MacKay, 2003] over the temporal states, thereby allowing for complex dependencies. Inference is performed via a double loop algorithm in which we optimize over the Gaussian process posterior in the outer loop and over temporal states in the inner loop. The resulting approximate posterior over the temporal states, $q(x_t|x_{t-1}, f)$, has a Markov Gaussian form and is found by a joint optimization over all temporal latent states. The latter addresses the issue of Ialongo et al. [2019] (which we also witness empirically), in which the parameters governing $x_t$ can only be learned once the estimate of the previous state, $x_{t-1}$, is meaningful. In the experiments we confirm the benefits of our novel inference scheme which provides higher quality uncertainty estimates than its state-of-the-art alternatives.

Our method is computationally efficient since we (i) can compute cheap gradients through the Laplace approximation by using the inverse function theorem [see e.g. Krantz and Parks, 2002], (ii) exploit the Markovian structure of our model [see e.g. Bell, 2000], (iii) approximate the Gaussian process posterior using inducing points [Quinonero-Candela and Rasmussen, 2005] and (iv) apply minibatching.

The remainder of this paper is structured as follows. In Sec. 2, we provide background, while we introduce our new method in Sec. 3. We relate it to existing work in Sec. 4, report experimental results in Sec. 5, and conclude in Sec. 6.

## 2 BACKGROUND

In this section we first provide background on the Laplace approximation for parametric state-space models and then on variational inference for Gaussian process state-space models before we briefly discuss general differences between the two approximate inference techniques.

### 2.1 STATE-SPACE MODELS AND THE LAPLACE APPROXIMATION

State-space models [see e.g. Särkkä, 2013] offer a principled way to model time series, i.e., noisy observations $Y_T = \{y_t\}_{t=1}^T$ from a dynamical system, where $y_t \in \mathbb{R}^{d_y}$. In order to disentangle the dynamics from the observational noise, state-space models use latent states $X_{T_0} = \{x_t\}_{t=0}^T$ with $x_t \in \mathbb{R}^{d_x}$ that are then assumed to form a Markov sequence.[1] Such a model is completely described by the *initial distribution* $p_\theta(x_0)$, a *transition model* $p_\theta(x_t|x_{t-1})$ and the *emission model* $p_\theta(y_t|x_t)$, resulting in

$$p_\theta(Y_T, X_{T_0}) = p_\theta(x_0) \prod_{t=1}^T p_\theta(y_t|x_t) p_\theta(x_t|x_{t-1}), \quad (1)$$

where generally the transition and emission model, and the initial distribution depend on model parameters $\theta$ that we wish to infer. Except for linear state-space models [Kalman, 1960], computing the marginal likelihood of the observations $Y_T$ under our model (also called evidence),

$$L(\theta) = p_\theta(Y_T) = \int p_\theta(Y_T, X_{T_0}) dX_{T_0}, \quad (2)$$

is not analytically possible and we have to resort to approximations. In the following, we introduce the Laplace approximation [see e.g. MacKay, 2003, Skaug and Fournier, 2006] which can be used to obtain an approximate maximum likelihood estimate of the parameters: Our goal is to find the setting of the parameters $\theta^*$ that maximizes the evidence, i.e., $\theta^* = \text{argmax}_\theta L(\theta)$. Defining $g(X_{T_0}, \theta) = \log p_\theta(Y_T, X_{T_0})$ (note that $Y_T$ is constant and hence not considered as a variable), we denote its maximizer[2] with respect to (wrt.) the latent states as $\hat{X}_{T_0} = \text{argmax}_{X_{T_0}} g(X_{T_0}, \theta)$. Performing a second order Taylor approximation of $g(X_{T_0}, \theta)$ around $\hat{X}_{T_0}$, plugging this back into Eq. (2), and then evaluating the resulting Gaussian integral yields (see Appx. B for a detailed derivation)

$$L(\theta) \approx \widetilde{p}_\theta(Y_T) \propto p_\theta(Y_T, \hat{X}_{T_0}) \det\left(H(\theta)\right)^{-\frac{1}{2}}, \quad (3)$$

$$H(\theta) = -\frac{\partial^2}{\partial X_{T_0}^2} g(X_{T_0}, \theta)\bigg|_{X_{T_0} = \hat{X}_{T_0}}. \quad (4)$$

Here and in the following we use $\widetilde{p}$ to denote a distribution, where the Laplace approximation has been applied to $X_{T_0}$ (and which therefore depends on $\hat{X}_{T_0}$). The expression in Eq. (3) can then be optimized numerically wrt. $\theta$ in order to estimate the parameters $\theta^*$ [Skaug and Fournier, 2006]. Note that this can be done efficiently since the Hessian $H$ in Eq. (4) is sparse and structured (see Sec. 3.3). The same methodology can also be applied efficiently to other latent variable models with a sparse or structured Hessian [e.g. Rue et al., 2009, Kristensen et al., 2016].

### 2.2 VARIATIONAL INFERENCE FOR GAUSSIAN PROCESS STATE-SPACE MODELS

Gaussian process state-space models [GPSSMs, see e.g. Frigola, 2015] are non-parametric extensions of the state-space models in Eq. (1). They employ a Gaussian process (GP) to model the possibly unknown dynamical transitions of the system. Next, we briefly describe GPs, how they can be used in state-space models, and how inference can be performed.

---

[1]We denote with capital letter with a capital index $(Y_T, X_{T_0})$ the collection of the respective lower case variables, and the index signifies the length of the collection $(Y_T = \{y_t\}_{t=1}^T)$. We use the index $T_0$ to denote the inclusion of $x_0$ in the collection $X_T$.

[2]This is in general a local optimum as the optimization problem is non-convex. However, if the observations are dense, the locally linear approximation to the dynamics made implicitly by the Laplace approximation is a reasonable assumption [see e.g. Eleftheriadis et al., 2017] leading to well-identifiable optima.

**Gaussian processes** A zero-mean GP $f \sim GP(0, k(\cdot,\cdot))$ is a distribution over functions and is fully specified by a positive-definite, symmetric kernel $k(\cdot,\cdot) : \mathbb{R}^{d_x} \times \mathbb{R}^{d_x} \to \mathbb{R}$. For every finite set of input points $X_M = \{x_m\}_{m=1}^M$, $x_m \in \mathbb{R}^{d_x}$, the outputs $F_M = \{f(x_m)\}_{m=1}^M$ are distributed according to a Gaussian distribution $p_\theta(F_M) = \mathcal{N}(F_M|0, K_{MM})$, where $K_{MM} = \{k(x_m, x_{m'})\}_{m,m'=1}^M$. Predictions for a new input point $x_t$ can be made by using the joint Gaussianity of $f_t \equiv f(x_t)$ with the $F_M$, and formulas for Gaussian conditionals resulting in $p(f_t|x_t, F_M) = \mathcal{N}(f_t|\mu(x_t, F_M), \Sigma(x_t))$, with mean and covariance

$$\mu(x_t, F_M) = K_{tM} K_{MM}^{-1} F_M, \tag{5}$$

$$\Sigma(x_t) = k_{tt} - K_{tM} K_{MM}^{-1} K_{tM}^\top. \tag{6}$$

Here $k_{tt} = k(x_t, x_t)$, and $K_{tM} = \{k(x_t, x_m)\}_{m=1}^M$. For a detailed introduction see Rasmussen and Williams [2006].

**Gaussian process state-space models** In a GPSSM, a GP prior is placed on the (mean of the) transition model that learns the mapping from a latent state $x_{t-1}$ to the next, $x_t$. Placing iid. Gaussian noise (with variance $Q$) on the transitions leads to

$$p_\theta(x_t|x_{t-1}, f_{t-1}) = \mathcal{N}(x_t|x_{t-1} + f_{t-1}, Q). \tag{7}$$

The resulting joint model is then given by

$$p_\theta(Y_T, X_{T_0}, F_T) = p_\theta(x_0) p_\theta(F_T|X_{T_0})$$
$$\times \prod_{t=1}^T p_\theta(y_t|x_t) p_\theta(x_t|x_{t-1}, f_{t-1}), \tag{8}$$

where $F_T = \{f(x_t)\}_{t=0}^{T-1}$. The emission model and the initial distribution remain unspecified since the methodology introduced in the following does not depend on them.

**Variational inference** Recent inference methods for GPSSMs [e.g. Doerr et al., 2018, Ialongo et al., 2019] rely on variational inference (VI). This approximate inference method works by choosing a parametric family of distributions $q_\psi(X_{T_0}, F_T)$ and optimizing the setting of the parameters $\psi$ such that $q$ is closest to the true posterior $p_\theta(X_{T_0}, F_T|Y_T)$ as measured by the Kullback-Leibler (KL) divergence $KL(q \parallel p)$. It can be shown that minimizing this KL divergence is equivalent to maximizing the so-called evidence lower bound (ELBO),

$$\mathcal{L}_{\text{VI}}(\psi, \theta) = \mathbb{E}_{q_\psi(X_{T_0}, F_T)} \log \frac{p_\theta(X_{T_0}, F_T, Y_T)}{q_\psi(X_{T_0}, F_T)}, \tag{9}$$

wrt. $\psi$ [for more details see Blei et al., 2016]. Furthermore, since the ELBO is a lower bound to the model evidence $\log p_\theta(Y_T)$ [cf. Eq. (2)], we can simultaneously learn the optimal model parameters $\theta^*$ by maximizing Eq. (9) wrt. $\theta$. These model parameters include for instance the hyperparameters of the GP (e.g. kernel length scales).

The different variational approximations to GPSSMs all have in common that they use sparse GPs [Snelson and Ghahramani, 2005, Titsias, 2009, Hensman et al., 2013] to model the GP part of the approximate posterior. The main idea of sparse GPs is to use a set of $M$ pseudo data points $\{X_M, F_M\}$ to summarize the information in the latent GP evaluations $F_T$. Here, the $X_M = \{x_m\}_{m=1}^M$ with $x_m \in \mathbb{R}^{d_x}$ are the inducing inputs that are placed in the same space as the latent states $x_t$, and the $F_M = \{f(x_m)\}_{m=1}^M$ are the so-called inducing outputs that share a joint Gaussian distribution, $p_\theta(F_T, F_M)$, with the $F_T$. The model by Doerr et al. [2018] that we will use in this work employs the fully independent training conditional (FITC) approximation [Snelson and Ghahramani, 2005] that assumes independence of the latent GP evaluations given the inducing outputs, i.e., $p_\theta(F_T|X_{T_0}, F_M) \approx \prod_{t=0}^{T-1} p_\theta(f_t|x_t, F_M)$. Notably, Doerr et al. [2018] also use this approximation in the augmented prior [cf. Eq. (8)], leading to $p_\theta(Y_T, X_{T_0}, F_M) = p_\theta(F_M) p_\theta(Y_T, X_{T_0}|F_M)$ with

$$p_\theta(Y_T, X_{T_0}|F_M) = p_\theta(x_0) \prod_{t=1}^T p_\theta(y_t|x_t) p_\theta(x_t|x_{t-1}, F_M). \tag{10}$$

Here we have already marginalized the $F_T$ from the prior model which can be done because of the FITC assumption:

$$p_\theta(x_t|x_{t-1}, F_M)$$
$$= \int p_\theta(x_t|x_{t-1}, f_{t-1}) p_\theta(f_{t-1}|x_{t-1}, F_M) df_{t-1}$$
$$= \mathcal{N}(x_t|x_{t-1} + \mu(x_{t-1}, F_M), Q + \Sigma(x_{t-1})), \tag{11}$$

where we used Eqs. (5)-(7) and standard formulas for Gaussian integrals. While the FITC assumption might seem quite restrictive and is criticized by Ialongo et al. [2019], we show in Appx. A that recent methods [including Ialongo et al., 2019] implicitly use this approximation as well, i.e., the methods could have used the approximation and would have arrived at the same optimization objective. This substantially weakens the criticism of Ialongo et al. [2019] on the choice of Doerr et al. [2018] on the FITC prior and motivates us to use it as well in our method. We discuss differences between these and other works in Sec. 4.

## 2.3 LAPLACE APPROXIMATION VERSUS VARIATIONAL INFERENCE

Before we finally present our proposed method that combines the approximate inference approaches presented in the previous sections we provide a short comparison of them. Using variational inference with a Gaussian variational family leads to a similar approach to the Laplace approximation since both methods use the same functional form for the approximate distribution and both approaches are mode seeking (see Fig. 1 left). While the Laplace approximation fits the mean to a mode of the distribution and has the same

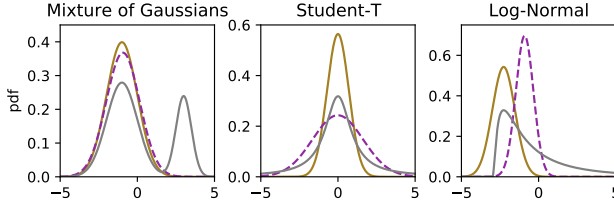

Figure 1: Qualitative difference between the Laplace approximation (brown, solid) and variational inference (magenta, dashed) for different ground-truth distributions (see subplot titles) depicted in gray.

curvature as the true function at this mode, variational inference minimizes the KL-divergence between the approximate and the true distribution. Due to the KL-divergence being heavily penalized by placing mass in regions that have zero mass under the true distribution, variational inference avoids this (see Fig. 1, right). Both approximations are not ideally suited to approximating heavy-tailed distributions due to their Gaussianity assumptions, but the Laplace approximation even slightly less so since matching the curvature at the mode typically leads to narrower distributions in this case (see Fig. 1, middle).

However, as we introduce next, in our model we only use the Laplace approximation for approximating the posterior over the temporal states. For many real-world applications, the dynamics can often be well described locally by a linear model justifying the Gaussian approximation [e.g. Eleftheriadis et al., 2017].

## 3 METHOD

In this section we propose a new inference method for the model used in Doerr et al. [2018], i.e. Eq. (10). Instead of relying solely on VI as previous works have done, we employ it in combination with the Laplace approximation. This allows us to treat the local latent variables, i.e. the temporal states $X_{T_0}$, and the global latent variables, i.e. the inducing outputs $F_M$, differently. Note that naively applying the Laplace approximation to this model, i.e., without making a distinction between the different sets of latent variables, (i) does not lead to an efficient algorithm since the resulting Hessian would not have an exploitable sparsity structure (see Sec. 3.3), (ii) would assume a linear relationship between the two latent variable classes and (iii) would make the Gaussian assumption more questionable.

We start by deriving our optimization objective in Sec. 3.1 and then, in Sec. 3.2, discuss a caveat that we encounter when trying to naively optimize this objective. Afterwards, we show how the sparsity of the Hessian in the Laplace approximation can be exploited for computational savings (Sec. 3.3) before summarizing the algorithm in Sec. 3.4.

### 3.1 COMBINING VARIATIONAL INFERENCE AND THE LAPLACE APPROXIMATION

Similarly as the previous methods, we wish to find an approximation to the log marginal likelihood $\log p_\theta(Y_T)$. We start from its definition given the model under consideration,

$$\log p_\theta(Y_T) = \log \int p_\theta(Y_T|F_M)p_\theta(F_M)dF_M, \quad (12)$$

where $p_\theta(Y_T|F_M) = \int p_\theta(Y_T, X_{T_0}|F_M)dX_{T_0}$. Using the Laplace approximation (Sec. 2.1) on this integral leads to

$$\log p_\theta(Y_T) \approx \log \int \widetilde{p}_\theta(Y_T|F_M)p_\theta(F_M)dF_M. \quad (13)$$

Here, $\widetilde{p}_\theta(Y_T|F_M)$ is given by [cf. Eq. (3)]

$$\widetilde{p}_\theta(Y_T|F_M) \propto p_\theta(Y_T, \hat{X}_{T_0}|F_M) \det\left(H(\theta, F_M)\right)^{-\frac{1}{2}}, \quad (14)$$

where $\hat{X}_{T_0}$ is a mode of the log-density

$$g_{\text{GP}}(X_{T_0}, \theta, F_M) = \log p_\theta(Y_T, X_{T_0}|F_M), \quad (15)$$

that can be evaluated using Eq. (10), and $H$ is the corresponding Hessian (cf. Eq. (4)). We proceed by using the VI methodology to lower bound the expression in Eq. (13): First, we multiply the term within the integral by $q_\psi(F_M)/q_\psi(F_M)$, where $q_\psi(F_M)$ is the variational distribution, an arbitrary distribution over the $F_M$ parameterized by $\psi$. Then, we use Jensen's inequality to push the logarithm inside of the resulting integral (thereby lower bounding the expression), and use the definition of the KL-divergence to arrive at our optimization objective,

$$\mathcal{L}(\theta, \psi) = \int q_\psi(F_M) \log \widetilde{p}_\theta(Y_T|F_M)dF_M$$
$$- KL(q_\psi(F_M) \| p_\theta(F_M)), \quad (16)$$

which obeys $\log p_\theta(Y_T) \gtrapprox \mathcal{L}(\theta, \psi)$. There are two things to note abound this bound: First, it is an approximate lower bound, since the Laplace approximation does not provide a valid bound but only an approximation. Second, the property that we minimize a KL-divergence between the true posterior $p_\theta(F_M|Y_T)$ and the approximate posterior $q_\psi(F_M)$ by optimizing this bound (see Sec. 2.2), also only holds approximately. In fact, the KL-divergence that is minimized by optimizing this bound is $KL\left[q_\psi(F_M) \| \widetilde{p}_\theta(F_M|Y_T)\right]$, i.e. $q_\psi(F_M)$ approximates the posterior after applying the Laplace approximation to the latent states $X_{T_0}$.

In principle, we could now go ahead, choose a parametric family for $q_\psi(F_M)$ and then evaluate our bound in Eq. (16) and use automatic differentiation to optimize the parameters $\theta$ and $\psi$. However, this is inefficient for two reasons: First, evaluating Eq. (14) involves an optimization to obtain $\hat{X}_{T_0}$ and automatic differentiation through this optimization is inefficient. Second, we need the determinant of the Hessian $H(\theta, F_M)$ in Eq. (14) which scales cubically in its size. In the following two subsections we provide solutions to both of these efficiency problems.

## 3.2 IMPLICIT FUNCTION THEOREM

Next, we turn to an important dependence in our construction: The mode $\hat{X}_{T_0}$ depends on the setting of the model parameters $\theta$. Since our optimization objective $\mathcal{L}(\theta, \psi)$ in Eq. (16) involves the mode $\hat{X}_{T_0}$ of the log-density $g_{\text{GP}}(X_{T_0}, \theta, F_M)$ [Eq. (15)], we require the derivative $\frac{\partial \hat{X}_{T_0}}{\partial \theta}$ in order to compute $\frac{\partial \mathcal{L}}{\partial \theta}$.[3] Being used to automatic differentiation we would usually leave this calculation to our favorite framework, since obtaining the mode $\hat{X}_{T_0}$ is nothing but a (rather long) sequence of summations and multiplications which automatic differentiation can deal with. Nevertheless, since several optimization steps are needed to compute the mode, backpropagating through optimization would lead to an enormous memory footprint and long execution times. Instead, we can calculate the derivative of the mode wrt. $\theta$ solely with the value of $\hat{X}_{T_0}$, i.e. independent of the steps taken to get there, with the help of the implicit function theorem [IFT, see e.g. Krantz and Parks [2002]].[4] In Appx. C, we show that the required derivative can be obtained as

$$\frac{\partial \hat{X}_{T_0}(\theta)}{\partial \theta} = H^{-1}(\theta, F_M) \frac{\partial h(\hat{X}_{T_0}, \theta, F_M)}{\partial \theta}, \qquad (17)$$

which we derive using the IFT and where $h$ [Eq. (42) in Appx. C] is the Jacobian of the function $g_{\text{GP}}$ [Eq. (15)] [see also Skaug and Fournier, 2006]. Both terms on the right hand side of Eq. (17) can be obtained using automatic differentiation and require only the value of $\hat{X}_{T_0}$ such that the complete computational graph of how it has been obtained is no longer required. Note that Eq. (17) exchanges potentially costly automatic differentiation computations with a Hessian solve. Naively, this would incur memory and time costs scaling quadratically and cubically in the size of the latent state, respectively. Therefore, this does only lead to an efficient algorithm by also exploiting the structure of the Hessian, as we will discuss next.

## 3.3 STRUCTURE OF THE HESSIAN

Taking a closer look at the definition of the Hessian $H(\theta, F_M)$ appearing in Eqs. (14) and (17), $H(\theta, F_M) = -\frac{\partial^2 g_{\text{GP}}(X_{T_0}, \theta, F_M)}{\partial X_{T_0}^2}$, we realize that it is given as the second partial derivatives of a sum of $T+1$ terms since $g_{\text{GP}}$ [Eq. (15)] is defined as the log-density of the distribution $p_\theta(Y_T, X_{T_0}|F_M)$ that consists of a product of $T+1$ terms [Eq. (10)]. Due to the Markovian structure of our model, all second partial derivatives wrt. latent states being more than one time step $t$ apart vanish, i.e., $H_{tt'}(\theta, F_M) =$

[3]Through the $F_M$, there is also a dependence on $\psi$ which must be treated in the same way. We omit the equivalent derivation for the sake of clarity and conciseness.

[4]The website implicit-layers-tutorial.org provides an introduction to the IFT geared to the Machine Learning practitioner.

$-\frac{\partial^2 g_{\text{GP}}(X_{T_0}, \theta, F_M)}{\partial x_t \partial x_{t'}} = 0$ for $t' \notin \{t-1, t, t+1\}$. This results in a block-tridiagonal structure of the Hessian:

$$H = \begin{pmatrix} A_0 & B_1 & 0 & \cdots & 0 \\ B_1^\top & A_1 & B_2 & \ddots & \vdots \\ 0 & B_2^\top & A_2 & B_3 & 0 \\ \vdots & \ddots & \ddots & \ddots & \vdots \\ 0 & \cdots & 0 & B_T^\top & A_T \end{pmatrix}, \qquad (18)$$

where $A_t = -\frac{\partial^2 g_{\text{GP}}}{\partial x_t \partial x_t}, B_t = -\frac{\partial^2 g_{\text{GP}}}{\partial x_t \partial x_{t-1}} \in \mathbb{R}^{d_x \times d_x}$ such that $H \in \mathbb{R}^{d_x(T+1) \times d_x(T+1)}$. This structure can also be found in similar models [see e.g. Bell, 2000] and the recent work by Durrande et al. [2019] considers the efficient implementation of computations for similar structures (banded matrices) into automatic differentiation frameworks.

The structure in Eq. (18) reveals another interesting aspect of our algorithm: The first step of the Laplace approximation consists of making a multivariate Gaussian approximation to the posterior $p_\theta(X_{T_0}|Y_T, F_M)$, where $\hat{X}_{T_0}$ is the mean and $H(\theta, F_M)$ is the precision matrix (see Appx. B, especially Eq. (40)). Therefore, the structure of $H$ tells us something about the underlying (implicit) structural assumption that we have used to approximate $p_\theta(X_{T_0}|Y_T, F_M)$. Exploiting the structure in Eq. (18) and using standard formulas for Gaussian conditionals, we can rewrite the approximate posterior to find a linear Markov Gaussian model,

$$\mathcal{N}\left(X_{T_0}|\hat{X}_{T_0}, H^{-1}\right) = \prod_{t=0}^{T} \mathcal{N}\left(x_t|a_t + b_t x_{t-1}, c_t\right),$$
$$(19)$$

where the coefficients $a_t$, $b_t$, and $c_t$ depend on $\hat{X}_{t-1:T}$ and the blocks $A_{t:T}$ and $B_{t:T}$ of the Hessian $H$ (here we used the shorthand $A_{t:T} = \{A_{t'}\}_{t'=t}^{T}$). Such a linear Markov Gaussian model is also used by Eleftheriadis et al. [2017], where the authors use parameters $a_t$, $b_t$, and $c_t$ that have to be optimized during inference while the conditional dependence on the $F_M$ is not taken into account.

There are three aspects of the algorithm for which we can achieve considerable computational savings when we take the structure of the Hessian into account: i) obtaining the blocks $A_t$ and $B_t$ [Eq. (18)] of the Hessian while not calculating the unnecessary zero blocks, ii) calculating the determinant of the Hessian required for Eq. (14), and finally iii) performing the Hessian solve in Eq. (17). Problem i), while not hard, requires a technical solution which we detail in Appx. D.1. Problems ii) and iii) can both be solved by following e.g. Koulaei and Toutounian [2007] in noting that the Hessian $H$ in Eq. (18) allows a factorization that can be exploited as we show in Appx. D.2. Implementing these improvements leads to a reduction of the memory footprint of the algorithm from $\mathcal{O}(T^2 d_x^2)$ to $\mathcal{O}(T d_x^2)$ [mainly through i)] and a reduction of the theoretical runtime from $\mathcal{O}(T^3 d_x^3)$ to $\mathcal{O}(T d_x^3)$ through ii) and iii).

## 3.4 ALGORITHM

In order to evaluate and optimize our optimization objective $\mathcal{L}(\theta, \psi)$ in Eq. (16), we need to choose a parametric family for $q_\psi(F_M)$. We follow the literature [e.g. Ialongo et al., 2019] and take a Gaussian distribution $q_\psi(F_M) = \mathcal{N}(F_M|m, S)$, allowing an analytical evaluation of the KL-term in Eq. (16). The first term on the right hand side of Eq. (16) is analytically intractable so we resort to sampling,

$$\int q_\psi(F_M) \log \widetilde{p}_\theta(Y_T|F_M) dF_M \approx \sum_{n=1}^{N} \log \widetilde{p}_\theta(Y_T|F_M^{(n)}),$$
(20)

with $F_M^{(n)} \sim q_\psi(F_M)$, and we use reparameterized samples [e.g. Kingma and Welling, 2014] to be able to compute derivatives wrt. $\psi$. The resulting basic algorithm to evaluate and optimize $\mathcal{L}(\theta, \psi)$ is summarized in Alg. 1 in Appx. E.

There are three extensions of this algorithm that are typically required for applying GPSSMs in practice (e.g. in Sec. 5.2), i) minibatches, ii) multi-dimensional latent states and iii) control inputs: Many time series are too long to be handled in one batch such that using i) minibatches helps obtaining a computationally tractable algorithm. For many real-world problems a one-dimensional latent state is not expressive enough [Frigola, 2015] and we require ii) multi-dimensional latent states $x_t$. Lastly, many datasets come with an additional time series $u_t \in \mathbb{R}^{d_u}$ of iii) external inputs that control the behavior of the system. We discuss the implementation of theses features into Alg. 1 in Appx. E.2.

## 4 RELATED WORK

There are two lines of work that directly relate to our approach: The first is on inference techniques for GPSSMs, the second on optimizing model parameters for latent variable models using the Laplace approximation.

**Inference for GPSSMs** The idea of using GPs to model transitions in state-space models goes back to Wang et al. [2005], where learning was performed by finding a maximum a posteriori estimate of the latent variables. The first Bayesian treatment of the latent states and the transition function in GPSSMs can be found in Frigola et al. [2013] using Markov Chain Monte Carlo. Due to the computational complexity of this approach, later works focused on variational approximations, beginning with Frigola et al. [2014]: They used a sparse GP in the flavor of Titsias [2009], introducing inducing outputs $F_M$ to deal with the transition function and using an independence assumption between the $F_M$ and the latent states $X_{T_0}$ in the variational posterior. This allowed them to find optimal variational distributions $q(F_M)$ and $q(X_{T_0})$ using variational calculus, where the latter is analytically intractable. This leads, similarly as in our work, to a double-loop algorithm, where in the inner loop

the distribution $q(X_{T_0})$ is approximated and in the outer loop $q(F_M)$ has to be obtained. The authors of Frigola et al. [2014] opt for a particle filtering method in the inner loop (note that the Laplace approximation would have also been possible here), but in contrast to our work do not take the conditional dependencies between the $F_M$ and the $X_{T_0}$ into account. Furthermore, this assumption allows for alternating updates for the parameters of $F_M$ and $X_{T_0}$ without having the need to differentiate through the latent states $X_{T_0}$.

Eleftheriadis et al. [2017] improved upon this method by using a doubly stochastic variational inference scheme that allows for the first time, and similarly as our approach, for minibatches. They opt for a parametric Gaussian distribution $q(F_M)$, a linear Markov Gaussian model for $q(X_{T_0})$ and additionally employ a recognition model to amortize the inference of the many parameters of such an approach. It is worth noting that our Laplace approximation (implicitly) also leads to a linear Markov Gaussian model as an approximation to the posterior $p(X_{T_0}|Y_T, F_M)$. However, our approach respects the conditional dependence of the temporal states on the inducing outputs $F_M$, and does not require any additional parameters to be learned since the means and variances of the Markov Gaussian model are obtained from the mode $\hat{X}_{T_0}$ and the Hessian $H$.

Recently, variational methods have also incorporated the dependence between the $F_M$ and the $X_{T_0}$ in their approximations: First Doerr et al. [2018], who have simply used the prior for $q(X_{T_0}|F_M)$ and then Ialongo et al. [2019], who employ a parametric non-linear Markov Gaussian model for the same term. A further subtle difference between these two approaches is that Doerr et al. [2018] use the FITC approximation from Snelson and Ghahramani [2005] in prior and approximate posterior, while Ialongo et al. [2019] do not use it and criticize its implications. However, it was already noted by Frigola et al. [2014], that the optimization objective for their model would not change if they applied the FITC approximation. We have confirmed in Appx. A, that this is also the case for the model of Ialongo et al. [2019], thus making this subtle difference even smaller.

Interestingly, even though the model of Doerr et al. [2018] is a special case of Ialongo et al. [2019], the latter work reports many cases in which the easier model still outperforms the more powerful one. This can potentially be attributed to the harder learning problem obtained through the additional introduction of the parameters of the Gauss-Markov model for $q(X_{T_0}|F_M)$ which only allows to learn the parameters of $q(x_t|x_{t-1}, F_M)$ after the preceding state $x_{t-1}$ has reached a meaningful state due to the sequential nature of the algorithm. In contrast to these two methods, we do not employ a variational distribution $q(X_{T_0}|F_M)$ but rather approximately marginalize $X_{T_0}|Y_T, F_M$ through the Laplace approximation.

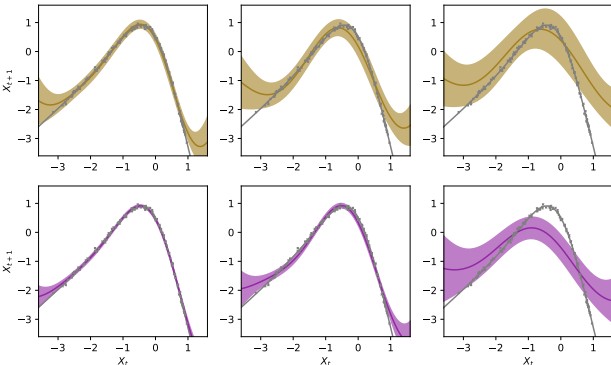

Figure 2: Sparse GP fits (mean $\pm\ 2\sigma$ confidence interval) to the *kink* transition function for our model (top) and VCDT (bottom) for varying emission noise ($\sigma_y^2 \in \{0.008, 0.08, 0.8\}$, left to right). Each plot shows the kink function (solid gray) and one sequence of $T = 120$ noisy latents $x_t$ drawn from that function (tiny gray crosses).

**Laplace approximation**   The other line of related work consists of approaches using the Laplace approximation for parameter estimation in latent variable models. One influential approach is Rue et al. [2009] that uses the Laplace approximation (twice) to perform approximate Bayesian inference of the model parameters in latent Gaussian models and pays close attention to the sparsity of the Hessian that is induced by different models. The work of Skaug and Fournier [2006] combines the Laplace approximation with automatic differentiation methods to arrive at an algorithm that can be used to approximate the marginal likelihood of (non-) Gaussian latent state models and therefore for maximum likelihood parameter estimation. The software package described in Kristensen et al. [2016] provides a recent implementation of the ideas in Skaug and Fournier [2006] with an additional focus on efficient automatic differentiation exploiting the sparsity of the Hessian. While closely connected to our work, these methods do not cover our approach since they jointly treat all latent variables, $\{F_M, X_{T_0}\}$, which would not lead to a Hessian whose structure can be exploited. Furthermore, a Laplace approximation over all latent variables would also lead to a less expressive approximation: This would entail a joint Gaussianity assumption of $F_M$ and $X_{T_0}$, while our current approach approximates both sets of variables with Gaussian distributions but allows for potentially complex and non-Gaussian interactions.

# 5   EXPERIMENTS

We first test the ability of our inference scheme to deal with different noise sources on the toy dataset called *kink* and then assess the performance of our method on a range of real-world benchmark datasets. Both experiments confirm that our new approach results in better calibrated predic-

Table 1: Comparison of our method with VCDT [Ialongo et al., 2019] on the *kink* data set. Shown are mean and standard errors over ten repetitions of the log-density (higher is better) of the *kink* function varying the emission noise variance $\sigma_y^2$. See Appx. F.1 for more details.

| MODEL | $\sigma_y^2 = 0.008$ | $\sigma_y^2 = 0.08$ | $\sigma_y^2 = 0.8$ |
|---|---|---|---|
| LAPLACE | 1.35(0.04) | **0.36(0.08)** | **-1.08(0.15)** |
| VCDT | 1.53(0.31) | -1.10(0.72) | -4.16(1.97) |

tions when compared with the Variationally Coupled Dynamics and Trajectories (VCDT) method from Ialongo et al. [2019] and the Probabilistic Recurrent State-Space Model (PRSSM) method from Doerr et al. [2018]. Comparisons to other time-series modeling approaches have already been performed in the latter work in which the PRSSM approach performed best. Hence, we do not repeat these experiments.

## 5.1   KINK

The *kink* function $f_k(x) = 0.8 + (x+0.2)[1 - 5/(1 + e^{-2x})]$ (Ialongo et al. [2019], see also Fig. 2 and Fig. 5 in Appx. G) can be used as a challenging transition function to probe state-space models: It tests the ability to model the nonlinear transition function and, by injecting additional noise, also the ability of the inference scheme to deal with different levels of emission noise. We generate data according to

$$x_t \sim \mathcal{N}\left(x_t | f_k(x_{t-1}), \sigma_x^2\right), \quad y_t \sim \mathcal{N}\left(y_t | x_t, \sigma_y^2\right),$$

for $t = 1, \ldots, T$, where we fix $T = 120$, $x_0 = 0.5$, $\sigma_x = 0.05$ and vary $\sigma_y^2 \in \{0.008, 0.08, 0.8\}$. Here, $\sigma_y^2 = 0.8$ corresponds to the setting of Ialongo et al. [2019] for which they empirically demonstrated that the inference scheme of Doerr et al. [2018] is not able to cope with the transition noise $\sigma_x$ and fails to learn the underlying dynamics. We follow Ialongo et al. [2019] and fix the emission model to the groundtruth, $p(y_t|x_t) = \mathcal{N}\left(y_t|x_t, \sigma_y^2\right)$. In addition, we choose a zero-mean sparse GP transition model with trainable Gaussian noise $Q$ [cf. Eq. (11) ], $p(x_t|x_{t-1}, F_M) = \mathcal{N}\left(x_t|\mu(x_{t-1}, F_M), Q + \Sigma(x_{t-1})\right)$ and a fixed initial distribution $p(x_0) = \mathcal{N}\left(x_0| -0.5, 1.5\right)$, all with one dimensional latent states $x_t$. We defer further details, the description of the setup of VCDT [Ialongo et al., 2019], and the initialization and training routines to Appx. F.1.

We present the resulting fits of the sparse GPs to the kink transition function and the noisy latent transitions in Fig. 2. We find that the GP in our model is well able to locate the kink in the *kink* transition function and finds better approximations with increasing signal to noise ratio (decreasing $\sigma_y^2$). The latter is also true for the VCDT method while the former does not hold for the largest $\sigma_y^2 = 0.8$. The small confidence intervals of the VCDT method sometimes result

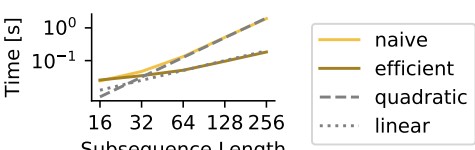

Figure 3: Timing comparison between a naive Laplace GPSSM and our proposed efficient implementation. The log-log plot shows the average time per training iteration as a function of the subsequence length $T$.

Table 2: Iteration number (divided by $100$) at which the mean parameters $A_t$ and $b_t$ of $q(x_t|\cdot)$ of the VCDT method [Ialongo et al., 2019] have converged for different time points $t$. Shown are mean and standard errors over ten repetitions using the *kink* dataset with $T = 120$ and $\sigma_y^2 = 0.08$.

|       | $t = 0$ | $t = 40$ | $t = 80$ | $t = 120$ |
|-------|---------|----------|----------|-----------|
| $A_t$ | 41(6)   | 53(5)    | 62(3)    | 68(4)     |
| $b_t$ | 53(5)   | 55(3)    | 67(4)    | 64(4)     |

in very good fits ($\sigma_y^2 = 0.008$), while other times leading to overconfident predictions (slightly for $\sigma_y^2 = 0.08$, evidently for $\sigma_y^2 = 0.8$), an observation that we have also made for the benchmark datasets in Sec. 5.2. In contrast, our method provides higher variance estimates resulting in very few data points not lying within the confidence intervals.

Next, we repeat the experiment 10 times, each time generating a new random dataset. Our observations also hold quantitatively as demonstrated in Tab. 1. In accordance with Fig. 2, we find that VCDT performs slightly, but not significantly, better than our method for $\sigma_y^2 = 0.008$, whereas we significantly outperform VCDT on the noisier data sets.

The controllable environment of the kink dataset also allows us to test if the theoretical speed-ups from Sec. 3.3 can be observed in practice. For this, we use the same setup as above, changing only $\sigma_y^2 = 0.01$ to provide an easy learning task and vary $T \in \{2^4, 2^5, 2^6, 2^7, 2^8\}$. We then compare the average runtime per iteration over the first 1000 iterations of our efficient implementation with a naive implementation, where all elements of the Hessian are calculated and the full Hessian is used to compute its determinant and the required Hessian solves, i.e., ignoring the improvements proposed in Sec. 3.3. The results of this comparison are depicted in Fig. 3 and clearly show that our efficient implementation scales linearly with the length of the time series $T$. For the naive implementation we observe a quadratic scaling with $T$ even though the theoretical scaling is $\mathcal{O}(T^3)$ (see Sec. 3.3). We attribute this to the huge cost of calculating the elements of the Hessian with automatic differentiation whose number scales quadratically for the naive version and linearly for our implementation. We hypothesize that the cubic scaling only sets in for very high values of $T$ which might become important if one wants to study long term effects requiring minibatches of increased size.

Finally, we also aim to validate our intuition that the parametric approach of Ialongo et al. [2019] for modeling the posterior over the latent states $q(x_t|\cdot)$ [see Eq. 51 in Appx. F.1] is problematic from a practical point of view. Their approach requires the sequential sampling of $x_t \sim q(x_t|\cdot)$ for $t = 1, \ldots, T$ during training which theoretically means that samples for $x_{t'}$ only become meaningful when the parameters of $q(x_t|\cdot)$ for all $t < t'$ have converged.

Our results in Tab. 2 support this thesis: There we show the average iteration number (out of 10000 in total) at which the variational parameters $A_t$ and $b_t$ of the mean of $q(x_t|\cdot)$ have converged when running the VCDT method with the same settings as for Tab. 1 on the *kink* data set (see Appx. F.1 for more details). There is a clear trend for $A_t$, and slightly less but still visible for $b_t$, that the variational parameters describing later time points $t$ also converge later during the optimization. We believe that this small experiment yields a possible explanation for why our approach, even though it uses a theoretically less expressive approximate posterior, outperforms the method of Ialongo et al. [2019].

## 5.2 SYSTEM IDENTIFICATION

We compare the performance of our method against PRSSM [Doerr et al., 2018] and VCDT [Ialongo et al., 2019] on five time series system identification benchmark datasets. Those consist of one dimensional time series of various lengths between 296 and 1024 data points and an equally long time series of one dimensional control inputs (see the appendix of Doerr et al. [2018] for more information about the datasets). For these more complicated tasks we choose a two dimensional latent state $x_t$ and a residual transition model with a sparse GP as in Eq. (11) with (diagonal) trainable Gaussian noise $Q$. We furthermore keep the initial distribution uninformative, $p(x_0) = \mathcal{N}(x_0|0, 1)$, but choose a slightly more expressive emission model [following Ialongo et al., 2019] $p(y_t|x_t) = \mathcal{N}(y_t|Cx_t + b, \Omega)$, where we fix $C = [1, 0]^\top$ and introduce trainable parameters $b$ and $\Omega$. Note that an even more expressive emission model does not lead to more expressivity of the composite model, only to more non-identifiabilities [Frigola, 2015]. For the PRSSM and VCDT methods, we use the original models detailed in Doerr et al. [2018] and Ialongo et al. [2019], respectively. For each dataset, we create ten different training tasks by varying the starting index of the training sequence, while keeping the length of the training sequence fixed to one half of the whole time series. Whereas Doerr et al. [2018] only compared the long-term predictions and Ialongo et al. [2019] only the short-term predictions, we evaluate both regimes by recording the predictive performance for varying time horizons in $T \in \{30, 60, 90, 120\}$. For more information

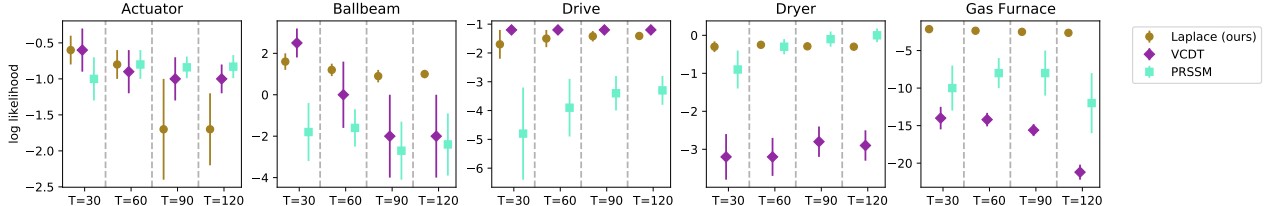

Figure 4: Comparison of average predictive log-likelihoods (higher is better) and their standard errors for ten repetitions on five different benchmark datasets. We evaluate our model (Laplace), VCDT [Ialongo et al., 2019] and PRSSM [Doerr et al., 2018] on predictions for $T \in \{30, 60, 90, 120\}$ steps in the future. See Appx. F.2 for more details.

about data splits, model configurations as well as training and prediction routines for each method, see Appx. F.2.

We plot the resulting test log-likelihoods in Fig. 4 (for a tabular comparison, see Tab. 3 in Appx. G). Our results clearly demonstrate that our method is a valuable addition to the set of inference methods for GPSSMs: For short-term predictions ($T = 30, 60$), our methods yields excellent results over all datasets, while VCDT shows deteriorated behavior on *Dryer* and *Gas Furnace*, and PRSSM on *Ballbeam*, *Drive* and *Gas Furnace*. For long-term predictions ($T = 90, 120$), our methods significantly outperforms its competitors on *Ballbeam* and *Gas Furnace*, while it underperforms on *Actuator*. We further report the root-mean-square errors (RMSE) in Tab. 4 in Appx. G. It is evident that a deteriorated log-likelihood value does not necessarily result in a large RMSE-value or vice versa. Instead, a drop in log-likelihood values is more likely caused by overconfident predictions which we can also witness in our exemplary plots in Fig. 6 in Appx. G.

In sum, we observed in our experiments that our method is able to learn the underlying dynamics for a variety of different tasksand outperforms its comparison partners. A direct attribution of our method's success is unfortunately not possible due to the nature of GPSSMs: They simultaneously learn the outputs of the sparse GP (through the inducing outputs) along with its inputs (the temporal states) such that a distinction between learning the two is impossible as their learning is inherently intertwined. This hinders a better theoretical understanding but also makes it very difficult to clearly attribute predictive improvements to certain parts of the learning process.

However, for a given GP, the Laplace approximation finds an optimal latent state at every iteration which we believe helps the GP convergence. In contrast, the fully variational approximation of Ialongo et al. [2019] performs an incremental update over the latent states in each optimization step which only leads to an optimal latent state after full convergence. It is therefore possible that the variational approximation of $q(x)$ for VCDT might be hindered by the optimizer getting stuck in a local optimum. Indeed, our empirical study indicates that VCDT is more suceptible to local optima than our method since the standard errors for VCDT are significantly higher in both experiments. While

this provides a plausible explanation, we cannot completely rule out other causes for this behavior.

Another potential reason for the success of our method is that the chosen variational family of Ialongo et al. [2019] is too compact which can result in narrow uncertainty estimates [e.g. Turner and Sahani, 2011]. Our experiments indicate that combining variational inference with the Laplace approximation favors less compact predictive distributions that lead to better calibrated predictions.

# 6  SUMMARY

In this paper, we have developed a new inference method for Gaussian process state-space models (GPSSMs) that combines a Laplace approximation over the temporal states $x_t$ with variational inference over the inducing outputs of the Gaussian process (GP) part of the model. Our approach learns a joint approximate posterior over the inducing outputs and the temporal states, refraining from sequentially learning the latter. We empirically find that our inference scheme is rewarded by better calibrated predictions compared to state-of-the-art methods.

While we only focused on the application of our inference scheme to GPSSMs, it can generally be applied to all models with two distinct sets of latent variables, e.g., the Bayesian treatment of model parameters in latent Gaussian models Rue et al. [2009] or of hyperparameters in (sparse) GP models [e.g. Hensman et al., 2013]. We further deem exchanging our variational inference engine with Hamiltonian Monte Carlo [Margossian et al., 2020] as an interesting avenue of future work on GPSSMs since recent research has shown promising results for a fully Bayesian treatment over GP hyperparameters and inducing locations [Rossi et al., 2021].

## Author Contributions

The initial problem setting and idea were given by Barbara Rakitsch while the details and experiments were designed by all authors jointly. Jakob Lindinger was responsible for working out the theoretical and technical details and implementing the algorithm. Jakob Lindinger and Barbara

Rakitsch performed the experiments, analyzed the results, and wrote the main paper with contributions of Christoph Lippert.

## Acknowledgements

The Bosch group is carbon neutral. Administration, manufacturing and research actvities do not longer leave a carbon footprint. This also includes GPU clusters on which the experiments have been performed.

We acknowledge constructive feedback from three anonymous reviewers that helped improve the paper and prompted us to perform the new experiment in Tab. 2. We furthermore acknowledge the helpful feedback from Sebastian Gerwinn and Çağatay Yıldız on a draft of the paper that greatly improved the motivation of this work.

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
