# OpenReview forum: "Laplace Approximated Gaussian Process State-Space Models"
_auai.org/UAI/2022/Conference — UAI 2022 Oral_

### Official Review · Reviewer_AZfS · 2022-04-10

**Q2(1) Originality/Novelty:** 3
**Q2(2) Significance/Impact:** 3
**Q2(3) Correctness/Technical Quality:** 3
**Q2(6) Clarity Of Writing:** 3
**Q6 Overall Score:** 7
**Q8 Confidence In Your Score:** 4

**Q1 Summary And Contributions:**

The paper presents a novel inference scheme for Gaussian Process State-Space models that applies stochastic variational inference for the
Gaussian process posterior and the Laplace approximation on the temporal states. The method is computationally efficient and leads to better calibrated predictions compared to state-of-the art alternative.

**Q2 Assessment Of The Paper:**

More detailed information regarding each of these aspects is given below:

**Q2(4) Quality Of Experiments (Optional):**

3: Good: The experimental evaluation is adequate, and the results convincingly support the main claims.

**Q2(5) Reproducibility:**

3: Good: Key resources (e.g., proofs, code, data) are available and key details (e.g., proofs, experimental setup) are sufficiently well-described for competent researchers to confidently reproduce the main results.

**Q3 Main Strengths:**

It is a very clear paper. The proposed solution combines well-known approaches, but the overall system is an improvement over previous work on GPSSM.

**Q4 Main Weakness:**

The Laplace approximation underestimates the covariance and tends to compute a mean close to the prior. However, it is used in combination with VI.

**Q5 Detailed Comments To The Authors:**

Good paper.

**Q7 Justification For Your Score:**

The proposed solution combines well-known approaches, but the overall system is an improvement over previous work on GPSSM. It is a simple but good idea.

**Q9 Complying With Reviewing Instructions:**

1: Yes.

---

### Official Review · Reviewer_eKP8 · 2022-04-11

**Q2(1) Originality/Novelty:** 3
**Q2(2) Significance/Impact:** 2
**Q2(3) Correctness/Technical Quality:** 3
**Q2(6) Clarity Of Writing:** 3
**Q6 Overall Score:** 6
**Q8 Confidence In Your Score:** 1

**Q1 Summary And Contributions:**

The paper introduces a novel method for inference in state-space models by considering a Laplace approximation on the transition model combined with a Gaussian process. The model makes clever use of the model structure and produces state-of-the-art results in experiments.

**Q2 Assessment Of The Paper:**

More detailed information regarding each of these aspects is given below:

**Q2(4) Quality Of Experiments (Optional):**

3: Good: The experimental evaluation is adequate, and the results convincingly support the main claims.

**Q2(5) Reproducibility:**

2: Fair: Key resources (e.g., proofs, code, data) are unavailable but key details (e.g., proof sketches, experimental setup) are sufficiently well-described for an expert to confidently reproduce the main results.

**Q3 Main Strengths:**

The paper is generally well written. It clearly states the problems the authors are trying to solve and how the proposed model solves them. The method makes intuitive sense and the authors have made use of the particular model structure, for example, when considering the structure of the Hessian to gain efficiency.

The experiments seem to support the claims made by the authors, delivering superior results, however, the code is not made available.

**Q4 Main Weakness:**

I don't know the relevant literature well enough to consider novelty or any serious technical flaws.

**Q5 Detailed Comments To The Authors:**

- The introductory paragraph is extremely general and doesn't add anything to the paper. I doubt anyone reading a paper from a conference on uncertainty would require a reminder that uncertainty exists. I recommend removing it.

**Q7 Justification For Your Score:**

The paper is generally well written and makes sense. The experiments seem convincing. While the research presented here does not fall into my area of expertise, I cannot find a reason for rejection.

**Q9 Complying With Reviewing Instructions:**

1: Yes.

---

### Official Review · Reviewer_ULgS · 2022-04-13

**Q2(1) Originality/Novelty:** 3
**Q2(2) Significance/Impact:** 2
**Q2(3) Correctness/Technical Quality:** 3
**Q2(6) Clarity Of Writing:** 4
**Q6 Overall Score:** 7
**Q8 Confidence In Your Score:** 3

**Q1 Summary And Contributions:**

This paper proposes using a Laplace approximation to improve inference in Gaussian process state-space models (GPSSMs). Specifically this approximation improves on the state-of-the-art by replacing sequential variational approximations of the latent states with a Laplace approximation. The paper address the computational challenge of obtaining the mode and Hessian required for the Laplace approximation. The propose scheme is shown to be better than existing methods in a pair of experiments.

**Q2 Assessment Of The Paper:**

More detailed information regarding each of these aspects is given below:

**Q2(4) Quality Of Experiments (Optional):**

3: Good: The experimental evaluation is adequate, and the results convincingly support the main claims.

**Q2(5) Reproducibility:**

3: Good: Key resources (e.g., proofs, code, data) are available and key details (e.g., proofs, experimental setup) are sufficiently well-described for competent researchers to confidently reproduce the main results.

**Q3 Main Strengths:**

The paper's main idea: replacing a sequential variational approximation with simpler-to-infer Laplace approximation is well motivated to help improve inference, without sacrificing modeling quality.

The paper proposes a clever solution to the naive computational challenges of calculating the mode and Hessian of the Laplace approximation by using IFT to compute the mode and exploiting conditional independence structure in the Hessian.

The paper's experiments support the main claim main contribution, showing their model out performs VCDT + PRSSM in both learning the kernel/latent transition function qualitatively on synthetic data (Sec 5.1) and quantitively on real data (Sec 5.2) .

The organization and presentation of the paper is well done. The background and related work is easy to follow and the appendix has relevant details including extensions of the model.

**Q4 Main Weakness:**

This is a minor point: how important are Gaussian Process State-Space Models in practice? The comparisons and experiments are against other state-of-the-art SSM inference methods, but does not consider other ML methods.

**Q5 Detailed Comments To The Authors:**

The subtle distinction between how a sequential variational approximation $q(x) = q_0(x_0) \prod_i q_i(x_i | x_{i-1})$ and the Laplace approximation impacts inference could be expanded. The paper could be made stronger if you could quantify the rate of convergence of $q_i(x_i | x_{i-1})$ for each $i$ to better motivate the intuition of needing to wait for $q_i'$ for $i' < i$ before infering $q_i$.

I'm curious in the attribution of your method success over the existing methods in the experiments. Does the success come from the Laplace approximation more quickly learning the correct kernel or latent approximation? Do the fully variational methods take longer to learn the kernel, are the variational approximation $q(x)$ getting stuck in local optima, etc.? Maybe some more elaborating on the last paragraph of section 5 could help.

Is the Laplace approximation necessary? Is there no way to apply the similar tricks in Section 3.2 and 3.3 to the sequential variational approximation $q(x) = q_0(x_0) \prod_i q_i(x_i | x_{i-1})$ by matching moments?

**Q7 Justification For Your Score:**

The work is technically solid and overcomes some nontrivial challenges to enable the Laplace approximation and the experiments advances the state-of-the-art in inference of GPSSMs.


**Q9 Complying With Reviewing Instructions:**

1: Yes.

---

### Decision · Program_Chairs · 2022-05-15

**Decision:**

Accept (Oral)

**Comment:**

Meta Review: The paper proposes a new inference method for Gaussian process state-space models  that combines a Laplace approximation over the temporal states  with variational inference over the inducing outputs. This approach learns a joint approximate posterior over the inducing outputs and the temporal states. It is empirically shown that the inference scheme leads to better calibrated predictions compared to state-of-the-art methods.

Rebuttal: the authors have carefully replied to reviewers' comments.

Suggested corrections: the authors should modify the paper as suggested by the reviewers. In particular, the revised version should include  the example they provided in the rebuttal "Distinction between sequential variational approximation and the Laplace approximation regarding". It clarifies the goal of the paper.